# Identification of *Candida* Species from Clinical Samples in a Honduran Tertiary Hospital

**DOI:** 10.3390/pathogens8040237

**Published:** 2019-11-15

**Authors:** Kathy Montes, Bryan Ortiz, Celeste Galindo, Isis Figueroa, Sharleen Braham, Gustavo Fontecha

**Affiliations:** 1Microbiology Research Institute, Universidad Nacional Autónoma de Honduras, Tegucigalpa 11101, Honduras; kathy.montes@unah.hn (K.M.); isis.figueroa@unah.edu.hn (I.F.); 2Instituto Hondureño de Seguridad Social, Tegucigalpa 11101, Honduras; 3Microbiology Research Institute, Universidad Nacional Autonoma de Honduras, Tegucigalpa 11101, Honduras; sharbraham@icloud.com

**Keywords:** *Candida* spp., PCR-RFLP, Honduras, HardyCHROM^®^, MicroScan^®^

## Abstract

*Candida* species are one of the most important causes of human infections, especially in hospitals and among immunocompromised patients. The correct and rapid etiological identification of yeast infections is important to provide adequate therapy, reduce mortality, and control outbreaks. In this study, *Candida* species were identified in patients with suspected fungal infection, and phenotypic and genotypic identification methods were compared. A total of 167 axenic fungal cultures and 46 clinical samples were analyzed by HardyCHROM^®^, MicroScan^®^(Omron Microscan Systems Inc, Renton, WA, USA), and PCR-RFLP (Restriction Fragment Length Polymorphisms). The species of the *C. albicans* complex were the most frequent, followed by *C. tropicalis* and *C. glabrata*. Less common but clinically relevant species of *Candida* were also isolated. The comparison between the three methods was concordant, especially for the most common *Candida* species. Fungal DNA amplification was successful in all clinical samples.

## 1. Introduction

Yeast infections vary in severity and can range from superficial infections [1] to sepsis with deadly potential [2]. One of the most relevant yeast genera due to its high morbidity and mortality is *Candida*. The *Candida* genus includes at least 15 species associated with human pathologies [3]. Most species of the genus *Candida* are considered normal inhabitants of the skin and mucous membranes. However, under specific circumstances, these microorganisms have the potential to express virulence factors that make them pathogenic or opportunistic, particularly in settings in which the patient suffers a debilitating clinical condition [4,5], or due to the use of invasive devices [6], or under immunosuppression [7]. The most severe candidiasis have a nosocomial origin and the source of infection can be both endogenous or exogenous [8].

*Candida albicans* is the most frequently reported species causing human infection [9], but other species are also reported: *C. glabrata*, *C. parapsilosis*, *C. tropicalis*, and *C. krusei,* among others [10]. In recent decades, the number of yeast infections seems to have increased significantly worldwide [11,12]. Although *Candida albicans* still remains responsible for most yeast infections, non-*albicans* species appear to be increasing in prevalence [13,14]. The greatest relevance of these recent changes in the distribution and epidemiology of severe *Candida* infections lies in the intrinsic differences between each of these species in their susceptibility to antifungal therapies [15]. In many cases, the identification of *Candida* species makes it possible to predict their potential susceptibility to antifungal medications.

In hospital laboratories, the routine identification of yeasts isolated from clinical samples is performed by phenotypic methods. In many low-income countries (LICs), microscopic observation of fungal structures in the clinical sample and culture are still considered the gold standard. Where available, the culture is usually followed by biochemical approaches based on chromogenic media for identifying the infecting species. Although these traditional methods are useful, they have some disadvantages, such as the prolonged time it takes to generate results until the identification of the microorganism is complete. Moreover, they have limited sensitivity and the interpretation of the results can be moderately subjective [16]. In countries with greater availability of financial resources, clinical laboratories have a greater number of techniques that allow rapid identification of yeast species, such as MALDI-TOF [17,18] or real-time PCR-based methods [19]. 

Whatever the diagnostic method used, it is very important to quickly identify the species of *Candida* responsible for an infection, in order to make timely decisions regarding appropriate therapy, to reduce mortality, control outbreaks, and carry out epidemiological investigations [20]. There are only two published studies regarding the distribution and frequency of isolated *Candida* species from clinical samples (urine and blood) from Honduras [21,22]. Consequently, epidemiological information on circulating *Candida* species in the country is scarce and insufficient. For this reason, the aim of this study was to detect and differentiate the *Candida* species in patients with suspected fungal infection in a third-level hospital in Honduras and compare the concordance between the traditional diagnostic phenotypic techniques used in Honduras with a molecular method based on PCR-RFLP (Restriction Fragment Length Polymorphisms).

## 2. Materials and Methods 

### 2.1. Clinical Samples and Yeast Isolation Conditions

A total of 167 axenic fungal cultures were prospectively obtained from the clinical laboratory of a tertiary-level hospital in Tegucigalpa, Honduras (Honduran Social Security Institute, IHSS) from January to August 2019. All positive cultures that showed a predominant presence of yeasts were included in the study. Yeasts were cultured from clinical samples that included urine (n = 63), sputum (n = 45), vaginal swabs (n = 18), blood (n = 12), catheters (n = 9), stool/rectal swabs (n = 5), cutaneous secretion (n = 2), otic secretion (n = 2), oral swabs (n = 4), cerebrospinal fluid (CSF) (n = 2), and abscesses (n = 5). These positive cultures were obtained from inoculating the clinical samples in blood agar or in chocolate agar plates, with the exception of blood samples that were cultured in specific culture bottles. The cultures were incubated at 37 °C for 24 to 48 h. The growth of microorganisms in blood cultures was monitored by the automated system BD BACTEC^TM^ FX (Becton, Dickinson and Company, NJ, USA) for 5 days. A subset of 46 biological samples used for yeasts isolation was separated for subsequent molecular analysis in order to detect and identify *Candida* species directly from the clinical samples: sputum (n = 13), vaginal swabs (n = 12), blood (n = 8), cutaneous secretion of the foot (n = 2), otic secretion (n = 2), oral swabs (n = 5), CSF (n = 2), and rectal swab (n = 2). Those samples were refrigerated until further processing. Some clinical samples, including all urine samples, were not used to directly detect yeast DNA due to the hospital’s working algorithm.

### 2.2. Phenotypic Identification of Yeast Species

Yeast species were phenotypically identified by two techniques. The first method was by culture of isolated colonies in HardyCHROM^®^ (CRITERION^®^, Hardy Diagnostics, Santa Maria, CA, USA). These media were incubated at 37 °C for 48 h and evaluated based on the color of the colonies according to the manufacturer’s instructions. A dark metallic green colony was interpreted as *C. albicans*; medium blue to dark metallic blue colonies, with a blue halo, were defined as *C. tropicalis*; pink to medium pink colonies were *C. krusei*; medium-sized, smooth, pink-colored colonies, often with a darker mauve center, were presumptively identified as *C. glabrata*; dry and dark purple colonies were assigned to *C. parapsilosis*, while the rest of species produced generally small, white- to pink-colored colonies. The second phenotypic method was the Rapid Yeast ID Panel in a MicroScan autoSCAN4^®^ (Siemens Healthcare, West Sacramento, CA, USA). One colony of the yeasts was taken from the HardyCHROM^®^ plate and the concentration of unit forming colonies was standardized using a 0.5 McFarland standard. This identification system is composed of microwells containing several identification substrates. This method yields results in 4 h and is able to identify 42 species of yeasts and 19 species of *Candida.*


### 2.3. In Silico Analysis of Restriction Fragments

Amplification sizes and restriction patterns of *Candida* species were calculated using the Geneious^®^ 9.1.7 software (Biomatters Ltd, Auckland, New Zealand). Sequences downloaded from the National Center for Biotechnology Information (NCBI) were trimmed to include target sequences for primers ITS1 and ITS4. The enzyme MspI was used for in silico digestion (Table 1).

### 2.4. DNA Extraction 

Cells of each culture were lysed with 1000 µL of a buffer composed of 10 mM Tris (pH 8), 1 mM Ethylenediaminetetraacetic acid (EDTA) (pH8), and 100 mM NaCl. This suspension was incubated in a water bath at 100 °C for 2 min and then stirred for 1.5 min at maximum velocity in a micro-mini BeadBeater^®^ system (Bio Spec products Inc., Bartlesville, OK, USA) with 0.5-mm glass beads. Supernatant was transferred to a 1.5-mL vial. One volume of phenol:chloroform (1:1) was added and mixed vigorously. After centrifugation at 13,000 rpm for 10 min, the aqueous phase was recovered and transferred to a new vial. Precipitation of nucleic acids was carried out by adding 1/10 volume of sodium acetate (3M, pH 5.2) and one volume of cold isopropanol. To facilitate the precipitation, samples were centrifugated at 13,000 rpm for 3 min. After careful removal of supernatant, the nucleic acids were washed three times with 300 µL of ethanol 70%. The dried pellets were suspended in nuclease-free water and stored at –20 °C until further use. The concentration of nucleic acids was calculated using a NanoDrop^®^ spectrophotometer (Thermo Fisher Scientific Inc., Waltham, MA, USA).

The DNA from the clinical samples was extracted following a protocol similar to that described above for yeast colonies with some modifications. These modifications included two initial washing steps of the sample in a buffer solution (Tris-HCI 10 mM, EDTA 0.1 M pH: 8) and 4 µL of proteinase k (20 mg/mL) during the lysis step, followed by incubation at 65 °C for 1 h. 

### 2.5. PCR-RFLP 

Molecular identification of *Candida* species was performed using a widely used method based on PCR-RFLP [21,23,24]. The amplification reaction was directed to the ribosomal region comprising the internal transcribed spacers ITS1 and ITS2 and the *5.8S* gene. Amplification conditions were carried out in a volume of 50 µL and included 25 µL of 2X PCR Master Mix (Promega Corp. Madison, WI, USA), 1 µL of each primer at a concentration of 10 µM, and 1 µL of DNA (40 ng/µL) as the template. The sequences of the two universal primers were: ITS1- 5´-TCC GTA GGT GAA CCT GCG G-3´, and ITS4- 5`-TCC TCC GCT TAT TGA TAT GC-3´. A Veriti™ 96-well Thermal Cycler (Thermo Fisher Scientific, Waltham, MA, USA) was used to amplify the DNA according to the following program: 95 °C for 5 min, 37 cycles of 95 °C for 30 s, 56 °C for 30 s, and 72 °C for 30 s, and a final extension at 72 °C for 5 min. Amplicons were visualized in 1.5% agarose gel electrophoresis with ethidium bromide. 

After confirming the amplification of the ribosomal region, 10 µL of the product were digested with the MspI enzyme at 37 °C for 2 h with 2 µL of buffer, 0.2 µL of 10 µg/µL acetylated Bovine serum albumin (BSA), and 0.5 µL of the restriction enzyme (10 U/µL) (Promega Corp., Madison, WI, USA). The digested fragments were analyzed on 2% agarose gel and recorded in a BioDoc-It Imaging System (UVP, LLC; Upland, CA, USA).

In order to ensure the integrity of DNA and an absence of the inhibitors from the clinical samples, a region of the human beta-globin gene was amplified as previously described [25] using the primers PCO3: 5´-ACA CAA CTG TGT TTC ACT AGC-3´ and PCO5: 5´-GAA ACC CAA GAG TCT TCT CT-3´.

### 2.6. Data Analysis

The Cohen´s kappa (k) coefficient, standard error (SE), and a 95% confidence interval were calculated to compare the agreement between the three methods (MicroScan^®^, HardyCHROM^®^, and PCR-RFLP). The molecular method was considered as the standard. In addition, the ability to detect *Candida* species directly from clinical samples was compared against the result of the culture.

### 2.7. Long-Term Preservation of Yeast Cultures

All strains were preserved under freezing at −20 °C in YPD medium (yeast extract, peptone, dextrose) and 99% sterile glycerol.

## 3. Results

### 3.1. Frequency of Candida Species According to HardyCHROM^®^

A total of 177 yeasts from 167 clinical samples were analyzed (Table 2). Phenotypic identification of yeasts was performed by chromogenic reaction in HardyCHROM^®^. In 10 cultures, two different species of yeasts (mixed culture) were obtained and identified separately. The clinical samples that most frequently showed yeasts as potentially responsible for infections were urine (37.7%), sputum (26.95%), and vaginal swab (10.78%). Eight different species of yeasts were identified, but one species could not be identified by HardyCHROM^®^. The most frequent species were *C. albicans* complex (42.93%), *C. tropicalis* (20.9%), and *C. glabrata* complex (16.94%).

### 3.2. Comparison of MicroScan^®^, HardyCHROM^®^, and PCR-RFLP

In order to assess the ability of two phenotypic methods commonly used in the hospital (IHSS) to identify yeast species, the results of both approaches were compared with a molecular technique (PCR-RFLP). Although most of the results are coincidental, there are some discrepancies between the three techniques (Table 3). When using the molecular method as a reference, the most common misidentifications of HardyCHROM^®^ were: Seven strains of *C. tropicalis* identified as *C. albicans*, one *C*. *glabrata* identified as *C. krusei*, and one *C. haemulonii* complex identified as *C. parapsilosis*. On the other hand, the most common errors of the MicroScan^®^ system were: Five strains of *C. glabrata* misidentified as *C. catenulata* (n = 1), *C. kefyr* (n = 1), *C. krusei* (n = 1), and two unidentifiable by PCR-RFLP. Likewise, nine *C. tropicalis* strains were incorrectly identified as *C. guillermondii* (n = 7), *C. catenulata* and *C. famata*. A strain of *C. haemulonii* complex was also misidentified as *C. famata*.

The kappa coefficient (k) was calculated to assess the level of agreement between the three methods (MicroScan^®^, HardyCHROM^®^, and PCR-RFLP). As shown in Table 4, the three methods have good levels of agreement (0.648–0.662), and there seems to be no significant differences between them.

### 3.3. Candida Species Detection and Identification Directly from Clinical Samples

In addition to assessing the ability of the three methods to correctly identify *Candida* species in axenic culture, DNA was extracted directly from 46 clinical samples. The ITS region of the yeasts was amplified through PCR and the amplicons were digested with MspI. It was possible to amplify the DNA of yeasts in all 46 clinical samples (100%), including those that are usually more difficult due to the presence of intrinsic inhibitors in the sample, such as feces and sputum. The most common species detected were *C. albicans* (41.2%), followed by *C. parapsilosis* (21.74%) and *C. tropicalis* (17.39%) (Table 5). 

When comparing the identification results of the isolated strains with those of the clinical samples, it was observed that in 34 cases (73.91%), the result was the same. However, a different species of *Candida* was identified in 12 (26.08%) clinical samples compared to that identified in the axenic culture (Table 6). Seven of these discrepancies came from samples of the oral cavity and respiratory tract, three from vaginal swabs, and one from skin.

## 4. Discussion

In the present study, we used three different methods (two phenotypic and one molecular) for the identification of *Candida* species isolated from 11 types of clinical samples from a tertiary hospital in Honduras. Yeasts isolated by HardyCHROM^®^ were further identified through PCR-RFLP. According to the molecular results, *C. albicans* complex was the most common species, followed by *C. tropicalis, C. glabrata* complex, *C. parapsilosis*, *C. krusei*, *C. kefyr, C. guillermondii*, and *C. haemulonii* complex. Although *C. albicans* was the predominant individual species, the rest of the non-albicans species contributed 57% of the total.

Several studies reported similar findings when analyzing clinical samples of diverse origins. A study conducted with clinical samples obtained from Iranian hospitals showed a higher frequency of *C. albicans*, followed by *C. parapsilosis*, *C. glabrata*, and *C. rugosa* identified by PCR-RFLP [26]. A second study analyzed five different types of clinical samples in Iran, and showed that *C. albicans* was the predominant species, followed by *C. glabrata* and *C. tropicalis* [27]. Another study conducted with 11 types of clinical samples from hospitalized patients with suspected fungal infection in Mexico City showed a higher prevalence of *C. albicans*, followed by *C. tropicalis* and *C. glabrata* [28]. In India, the identification by PCR-RFLP of 150 strains from different clinical samples also showed a majority of *Candida albicans*, followed by *C. glabrata*, *C. tropicalis*, and *C. parapsilosis*. Kaur et al. analyzed *Candida* strains isolated from six types of clinical samples, and also reported an important predominance of *C. albicans* [29]. Other studies have analyzed a single type of clinical sample, such as blood [30,31,32,33], urine [21,24], vulvovaginal secretion [34], bronchoalveolar fluid [35], and gastroesophageal tract [36]; however, in all of them the predominant species was *C. albicans*. Although there are many epidemiological studies that show a trend towards an increase in infections caused by non-albicans *Candida* species [37], it seems that the *C. albicans* complex remains the most frequent etiological agent among fungal human infections when considered individually. However, when non-albicans species are considered as a group, these are the majority with respect to *C. albicans.*

According to our results, the species most commonly isolated from blood was not *C. albicans* but *C. parapsilosis* (n = 5; 41.7%). This result is interesting because *C. parapsilosis* complex species have emerged as an increasing cause of fungemia [38], and due to its capacity as a skin colonizer that facilitates transmission from health personnel to patients during the manipulation of intravascular catheters [39]. In urine, however, the most frequent species were *C. albicans* complex, *C. tropicalis*, and *C. glabrata* complex. A previous study in search of the primary cause of genitourinary infection in a hospital in Honduras analyzed 73 yeasts, with the results consistent with the current study [21]. 

The second objective of this study was to compare the capacity of two biochemical approaches and a molecular method to identify *Candida* species. Most species identifications among the three methods coincided and showed good kappa agreement, especially for the most common species. Many studies have also shown high agreement between molecular findings and traditional phenotypic tests, such as chromogenic media and automated biochemical approaches [24,29,31,34,35,40,41]. Consequently, the gold standard for the diagnosis of *Candida* infections based on biochemical methods seems to be specific enough for a routine hospital diagnosis. 

However, when considering PCR-RFLP as a reference standard in this study, we found that HardyCHROM^®^ and MicroScan^®^ failed to correctly identify 9 (5.08%) and 15 (8.47%) isolates, respectively. The three species misidentified by these methods were *C. tropicalis*, *C. glabrata* complex, and *C. haemulonii* complex. An important drawback of the phenotypic methods evaluated in this study is its inability to correctly identify species of the *C. haemulonii* complex, to which *C. auris* belongs. *C. auris* is an important emerging pathogen with a high mortality rate in hospitals [42].

There are several other reports of inconsistency between molecular and phenotypic assays. Zhai et al. [34] compared the results of a molecular method based on real-time PCR with CHROMagar Candida^®^, and detected that 9.3% of inconsistencies were mostly attributable to the phenotypic method. A study in which several methods were compared, including a chromogenic medium and PCR-RFLP, determined a 2.5% inconsistency between the assays [31]. Jafari et al. compared CHROMagar Candida^®^ and two molecular assays. According to their results, the concordances (k coefficient) between the phenotypic assay and the molecular methods were 0.87 and 0.89 [26]. 

The causes of inconsistencies between methods could be attributed to a limited ability of phenotypic approaches, such as HardyCHROM^®^, to identify rare species [26], nor can PCR-RFLP identify all potential restriction patterns. Another reason could be the presence of more than one species of yeast in a seemingly pure culture. These mixed cultures have been described frequently [21,31,43,44], and in cases in which one of the species is underrepresented, they may cause confusion in the interpretation of biochemical results. On the other hand, the use of two phenotypic methods can be considered a good practice that provides greater specificity and sensitivity in the identification of species, mainly when there are two or more species co-infecting a patient. 

Currently, molecular techniques based on PCR are not recommended nor have they been approved for clinical diagnosis purposes [45]. Despite this, the third objective of this study was to identify the yeast species by directly extracting the DNA from eight types of clinical samples. Indeed, it was possible to amplify *Candida* DNA in all of the 46 samples analyzed. The species identified in the clinical samples mostly coincided with the results of the molecular identification from the cultures (73.91%). In 12 cases, the results were discordant, especially in samples taken from mucous membranes. Since the mucous membranes are normally colonized by commensal yeast species [46] it is possible that in some cases, the species responsible for the infection shares an ecological niche with other species and this mixture could be responsible for the discrepancies in the diagnosis. In the sterile clinical samples, the results were more satisfactory. In the eight blood samples analyzed, seven revealed the same species, and only one sample identified *C. tropicalis* while the culture identified *C. guillermondii*. In the two CSF samples, the results were concordant. This last result is interesting since an early, sensitive, and specific diagnosis of the etiologic agent in a sterile sample, independent of the culture, could positively influence the patient’s outcome and survival.

## 5. Conclusions

In this study, *Candida* species isolated from clinical samples of a third-level hospital in Honduras were described. The most frequent species were *C. albicans* complex, *C. tropicalis*, and *C. glabrata* complex. The capacity of two biochemical methods and a molecular assay for the correct identification of *Candida* species was compared, showing good agreement between the three methods. However, the phenotypic methods are disadvantaged in that they are unable to correctly identify some uncommon species. Finally, we determined that it is possible to identify *Candida* species with relative success through PCR-RFLP from clinical samples, especially from blood and CSF.

## Figures and Tables

**Table 1 pathogens-08-00237-t001:** Distinctive restriction fragments for *Candida* spp. produced by the enzyme MspI on the ITS1–ITS2 region.

*Candida* Species	Length of the ITS1-ITS2 Amplicon (bp)	Restriction Fragment Sizes (bp)
*C. albicans* complex	538	299, 239
*[Candida] ^1^ glabrata* complex	880	563, 317
*C. parapsilosis* complex	520	520
*C. tropicalis*	528	342, 186
*C. krusei* (*Pichia kudriavzevii*)	510	262, 248
*C. kefyr* (*Kluyveromyces marxianus*)	721	721
*C. guillermondii* (*Meyerozyma guilliermondii*)	607	372, 157, 82
*[Candida] haemulonii* complex	400	400
*C. catenulata* (*Diutina catenulata*)	402	402
*C. famata* (*Debaryomyces hansenii*)	639	639
*[Candida] zeylanoides*	626	626
*[Candida] inconspicua*	455	245, 210

^1^ Square brackets ([]) around a genus indicates that the name awaits appropriate action by the research community to be transferred to another genus, according to NCBI.

**Table 2 pathogens-08-00237-t002:** Number of *Candida* species isolated in HardyCHROM^®^ agar and identified through PCR-RFLP.

Clinical Sample	N° of samples (%)	*C. albicans* Complex	*C. glabrata* Complex	*C. parapsilosis* Complex	*C. tropicalis*	*C. krusei*	*C. kefyr*	*C. haemulonii* Complex	*C. guillermondii*	Unidentified	Total n° of Yeasts (%)
Urine	63 (37.70)	25	16	1	20	1	2				65 ^1^
Sputum	45 (26.95)	28	6	3	10	1	2			1	51 ^1^
Vaginal swab	18 (10.78)	11	5		1	1	1	1			20 ^1^
Blood	12 (7.18)	2		5	1	2			2		12
Catheter	9 (5.39)	3	2	3	1						9
Stool	5 (2.99)			2	1	2					5
Cutaneous secretion	2 (1.20)	1			1						2
Otic secretion	2 (1.20)			1		1					2
Oral swab	4 (2.39)	3			1						4
CSF	2 (1.20)	2									2
Abscess	5 (2.99)	1	1	1	1			1			5
Total (%)	167 (100)	76 (42.93%)	30 (16.94%)	16 (9.03%)	37 (20.9%)	8 (4.51%)	5 (2.82%)	2 (1.12%)	2 (1.12%)	1 (0.56%)	177 (100%)

^1^ Clinical samples with mixed cultures including two or three different colonies.

**Table 3 pathogens-08-00237-t003:** Number of *Candida* species isolated in axenic culture and identified by two phenotypic methods and a molecular technique.

*Candida* species	MicroScan^®^	HardyCHROM^®^	PCR-RFLP
*C. albicans* complex	69	74	76
*[Candida] glabrata* complex	25	29	29
*C. parapsilosis* complex	11	10	16
*C. tropicalis*	26	32	37
*C. krusei*	10	4	8
*C. guillermondii*	11		2
*[Candida] haemulonii* complex			2
*Candida kefyr*	4		5
*C. famata*	6		
*C. catenulata*	4		
*[Candida] inconspicua*	1		
*[Candida] zeylanoides*	1		
Total	168	149	175

**Table 4 pathogens-08-00237-t004:** Cohen’s kappa coefficient between two phenotypic methods and PCR-RFLP for the identification of *Candida* species.

Method	PCR-RFLP	HardyCHROM^®^	MicroScan^®^
MicroScan^®^	0.648 (0.041; 0.568–0.727) ^1^		
PCR-RFLP		0.653 (0.042; 0.572–0.735) ^1^	
HardyCHROM^®^			0.662 (0.039; 0.568–0.739) ^1^

^1^ SE of kappa; 95% confidence interval.

**Table 5 pathogens-08-00237-t005:** Number of *Candida* species identified directly from the clinical sample through PCR-RFLP.

Clinical Sample	N of Clinical Samples (%)	*C. albicans* Complex	*C. glabrata* Complex	*C. parapsilosis* Complex	*C. tropicalis*	*C. krusei*	*C. haemulonii* Complex
Sputum	13 (28.26)	8	1	1	3		
Vaginal swab	12 (26.09)	5	3		2	1	1
Blood	8 (17.39)	2		4	2		
Cutaneous secretion	2 (4.35)		1	1			
Otic secretion	2 (4.35)			1		1	
Oral swab	5 (10.87)	2		1	1		1
CSF	2 (4.35)	2					
Rectal swab	2 (4.35)			2			
Total	46 (100)	19 (41.3)	5 (10.87)	10 (21.74)	8 (17.39)	2 (4.35)	2 (4.35)

**Table 6 pathogens-08-00237-t006:** Discordant identification of *Candida* species between axenic cultures and clinical samples through PCR-RFLP.

Clinical Samples	Axenic Culture	*n*
*C. tropicalis*	*C. albicans* complex	2
*C. parapsilosis* complex	*C. albicans* complex	2
*C. albicans* complex	*C. tropicalis*	3
*C. albicans* complex	*Candida* spp.	1
*C. tropicalis*	*C. guillermondii*	1
*C. albicans* complex	*C. krusei*	1
*C. haemulonii* complex	*C. kefyr*	1
*C. glabrata* complex	*C. haemulonii* complex	1

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
