# Peer review of "Identification of Candida Species from Clinical Samples in a Honduran Tertiary Hospital"

_pathogens, 2019, doi:10.3390/pathogens8040237_

Round 1

Reviewer 1 Report

 This article provides a quantitative comparison between existing methods for identifying Candida species. The authors test the efficacy of these methods to identify different clinically relevant Candida species from axenic culture and clinical specimens without culturing. The results described has the potential for guidelines for what conditions the methods are effective. Overall this article was well written and very interesting.

It is interesting that axenic samples frequently exhibited yeast originated from urine. However, the authors do not demonstrate if they can detect yeast from urine samples without culturing. Have authors done this comparison or can they comment on why it was not included?

Also there are different cutaneous microbial environments. The authors do not describe the location, on the body of cutaneous secretion samples were obtained.

Table 2 the components of the table are misaligned.

Line 69 it is not clear how positivity was determined.

Line 120 "de" should be deleted.

Line 167-171 it is unclear which method is being reported or which samples are being compared by PCR/RFLP. The incorrectly identified strains were by which method (line 169-171).

Author Response

It is interesting that axenic samples frequently exhibited yeast originated from urine. However, the authors do not demonstrate if they can detect yeast from urine samples without culturing. Have authors done this comparison or can they comment on why it was not included? A. The authors greatly appreciate the reviewer's contributions to improve the manuscript. In relation to the question, in fact, in this study it was not possible to perform molecular analyzes directly from urine samples due to a logistical problem. In the case of urine, the samples are discarded after being used for microbiological culture. Since the study included only positive cultures with yeasts, and these cultures became positive 24 hours after culture, samples were no longer available at the time of inclusion in the study. Although we cannot show this relevant information in this study, we do not see any reason why it is not possible to amplify the yeast genome directly from this clinical sample. To clarify this idea, the following sentence has been included in section 2.1 of Material and Methods: “Some clinical samples, including all urine samples, were not used to directly detect yeast DNA due to the hospital's working algorithm”. Also there are different cutaneous microbial environments. The authors do not describe the location, on the body of cutaneous secretion samples were obtained. A. The reviewer is correct. Indicating the origin of the sample is highly relevant. The two cutaneous secretions were obtained from the feet of the patients. In the manuscript we have indicated this in section 2.1 of Material and Methods. Table 2 the components of the table are misaligned. A. The reviewer is right. The components of some tables are misaligned due to their large size; however, we are sure that this problem will be solved during the document format phase before publication. To solve this problem, the authors cannot do anything about it. Line 69 it is not clear how positivity was determined. A. The positivity of the cultures from clinical samples is determined by two criteria: (1) the presence or absence of microorganisms with morphology characteristic of the expected pathogens in that clinical sample; and (2) in the case that the cultures show the presence of two or more morphotypes, the predominance criterion is also important. Line 120 "de" should be deleted. A. The paragraph has been modified as follow: “These modifications included two initial washing steps of the sample in a buffer solution (Tris-HCI 10 mM, EDTA 0,1M pH: 8) and 4 µL of proteinase k (20 mg/mL) during the lysis step, followed by incubation at 65 ℃ for 1 hour.” Line 167-171 it is unclear which method is being reported or which samples are being compared by PCR/RFLP. The incorrectly identified strains were by which method (line 169-171). A. In this paragraph two phenotypic methods and one molecular method (PCR-RFLP) are compared. It is indicated that the method used as a reference is the molecular method. Then, the number of species that were misidentified by HardyCHROM is indicated, and then the number of species that were misidentified by MicroScan is also indicated. The authors fail to find the reason why the paragraph is unclear. 

Reviewer 2 Report

This is a straightforward study that compared three different methods of identification of Candida species recovered from multiple human body sites. Although restricted to a specific clinical population in Honduras, this study provides a thorough examination of the efficacy of three different methods to differentiate Candida species. The Tables are easy to read and appropriate, and the conclusions justified. I was surprised by the high reliability of simple HardyCHROM culture for identification, although the mis-identifications were also interesting. The only concern I have is the use of the term "complex" along with the species- I don't understand what this refers to.

Overall a simple but valuable study.    

Author Response

This is a straightforward study that compared three different methods of identification of Candida species recovered from multiple human body sites. Although restricted to a specific clinical population in Honduras, this study provides a thorough examination of the efficacy of three different methods to differentiate Candida species. The Tables are easy to read and appropriate, and the conclusions justified. I was surprised by the high reliability of simple HardyCHROM culture for identification, although the mis-identifications were also interesting. The only concern I have is the use of the term "complex" along with the species- I don't understand what this refers to. Overall a simple but valuable study.  A. The authors greatly appreciate the reviewer's comments in order to improve the manuscript. Indeed, we have decided to use the term "complex" in the case of four species (C. albicans, C. glabrata, C. parapsilosis, and C. haemulonii) because there are 3 to 4 different species in each of these complexes that cannot be differentiated by their phenotype or by PCR-RFLP. Consequently, in order to be strict, this study cannot accurately indicate the identity of the species within these 4 complexes.